

# Analysis of Land surface Temperature change based on MODIS data, Case study: Inner Delta of Niger

Dembélé Abdramane [1,3], Xiufen Ye [1], Touré Amadou [2]

[1]Automation, Harbin Engineering University, Harbin 150001, Heilongjiang, China
[2]forestry, Northeast Forestry University, Harbin, 150040, Xiangfang District, China
[3]Geodesy, National School of Engineers ENI (Ecole Nationale d'Ingenieurs-Mali), 410, Av Vollenhoven, 242, Bamako, Mali

*Correspondence to*: Dembélé Abdramane (a.dembele10@yahoo.fr)

**Abstract.** Land Surface Temperature (LST) investigation in the Sahel zones is a crucial task to counter the climate change effects. Inner Delta of Niger (IDN) affected by a significant change of LST over an eighteen years period (from 2000 to 2017)
is threatened by natural risks like volcanic hazards and the degradation of the global environment. This work focuses on the Early Warning Systems and Monitoring Technologies of the LST change over the existent phenomena and different types of geologies. Indeed, the processing of MODIS (Moderate-Resolution Imaging Spectroradiometer) data was carried out from Geographic Information System (GIS) and remote sensing (RS) methods including "equal interval" method followed by the highlighting of hottest sectors as well as their delineations. The diachronic analysis of processed images into 5 temperature
slices at equal interval, shows a temperature increase over the said period with an annual rate of increasing temperatures of 0.24°C. The spatiotemporal dynamics of temperature slices "19.21°C to 25.15°C", "25.15°C to 31.10°C" and "31.10°C to 37.05°C" shows an extension of surfaces with mean annual progress rates of  0.13%, 0.20%, and 1.74% respectively. At the same time, mean annual regress rates of -0.64%, and -1.42% has been observed at the temperature slices "37.05°C to 43.00°C" and "43.00°C to 48.94°C". The attrition (disappearance of spots) of 12.22% represents the dominant spatial transformation
process of the maximum temperature slice "43.00 °C to 48.94 °C" which spreads over a mean surface of 4.42%. Thus the maximal temperature increases while its occupancy surface decreases. Therefore, The IDN threatened by the desertification is affected by a strong terrestrial global warming determining the volcanic hazards areas (Faguibine Lake).

## 1 Introduction

The MODIS image is a good indicator of Land surface temperature (LST) on the interface analysis in order to characterize
areas. The latest decades has registered the attraction of much attention through the LST at large-scale (Roy et al., 2014; Vlassova, Pérez-Cabello, Mimbrero, Llovería, & García-Martín, 2014; Wang, Liang, & Meyers, 2008; Zorer et al., 2013) and the advancement of earth and environmental sciences in the desertification monitoring so as in the monitoring of surfaces at the volcanic hazards (Hillger & Clark, 2002a, 2002b). One of the major driving forces causing many extremes of soil anomalies whose in recent years that many regions have undergone is the terrestrial global warming (C et al., 2013; Committee, 2015;
Tol, 2009). Based on meteorological stations data, many studies have quantified the global temperature of the earth (Coumou,



Robinson, & Rahmstorf, 2012; Frangou, Ladle, Malhado, & Whittaker, 2010; Rahmstorf & Coumou, 2011). Then several phenomena and climate extreme have been recorded during 2011–2015 and the probability increased by a factor of ten or more, in the case of some extreme high temperatures (World Meteorological Organization, 2016a)(World Meteorological Organization, 2016b). Only temperature anomalies have concerned most analysis(Report, 2018; Weber et al., 2014). The Inner

Delta of Niger (IDN), in central of Mali, through desert regions represents one of the hot areas of the African plate (Direction, 2008; Mao, K et al., n.d.). Inner Delta of Niger, is part of a large geological structure (Continental Terminal, Precambrian, Quaternary) that have played several roles since the Pan-African Orogeny and may have current activity (Inger, Dione, Jarosewich-Holder, & Olivry, 2006), (El Abbass et al., 1993). Certain portions of the region are superimposed on an area of positive gravity anomalies (Svensen et al., 2003). The parallelism between the gravity and structural directions is particularly

marked between Tombouctou, Faguibine and the Nara Trench (Chudeau, 2018; Zwarts, Van Beukering, Kone, & Wymenga, 2005). In April 2001, has been receiving reports about increased thermal activity in Tombouctou area (Svensen et al., 2003). Hot fumaroles and magmatic rocks whose the upper part of a hydrothermal system has been observed and considered as volcanic activity (Villatte, 1973). The objective of this work is to analyze temporally the LST of the Inner Delta of Niger in relation to the different sectors, then to document the evolution of the surface warming.

**2 Study area**

Inner Delta of Niger is located in the central part of Mali in West Africa (figure 1) and it is delimited by longitudes 6.203597°W, 1.492470°W and latitude 13.318241°N, 17.340885°N, covering an area of approximately 131886.607 km$^2$. The Niger River crosses the study area from southwest to the northeast direction. Its geology is composed at the north by the dunes and sandy, Clays, laterite; in the Tombouctou region (Dembélé & Ye, 2017; Olivry, 2002), where are located the Faguibine lake, lake

Kamaga, and lake Gouber with a surface temperature as high as 765 °C in January 2002 expressing a volcanic activity (Irrigués, Piv, & Takara, 2007; Villatte, 1973). A movement of air masses from the Intertropical Convergence Zone is associated  with weather conditions (Inger et al., 2006; WWF, 2006).



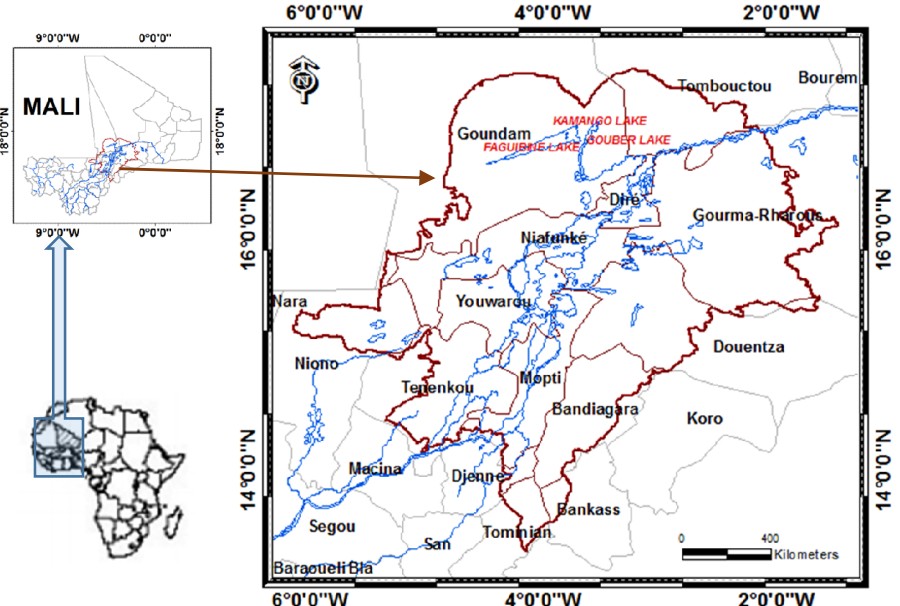

**Figure 1.** Study area location

# 3 Methods and Data

The developed methodology shows different steps to derive temperature series from MODIS images (figure 2). The archived images have been selected from databases of the website "earth explorer" in the way that the study area must be completely
5  covered by a good visualization until 95% (Survey, 2005; USGS United States Geological Survey, 2016; Wan, 2007). The unit of the generated temperature of the land surface by the Landsat-8 satellite is in Kelvin degree with 0.928 kilometer of spatial resolution. Characteristics of the effective calibration parameters of Scientific Data Sets (SDS) are expressed in the below Table A (Wan, 2007). The used images MODIS MOD11A1V6 are selected over a period of eighteen years (2000 to 2017). The used method presents 3 separate parts whose preprocessing includes the MODIS data clipping with the study area
10  followed by the data screening for Kelvin degree and the conversion in Celsius degree. The "equal Interval" method has allowed processing each image into 5 classes of temperature, then to highlight the extent of different surfaces having high temperatures (figure 3). The study area divided between different and geologies and landscapes formations established an LST dynamics and an LST modelling (figure 3 and figure 6). Thereby, analyze LST transition, STP evolution of LST, and area morphology to characterize the different sectors of the study area.



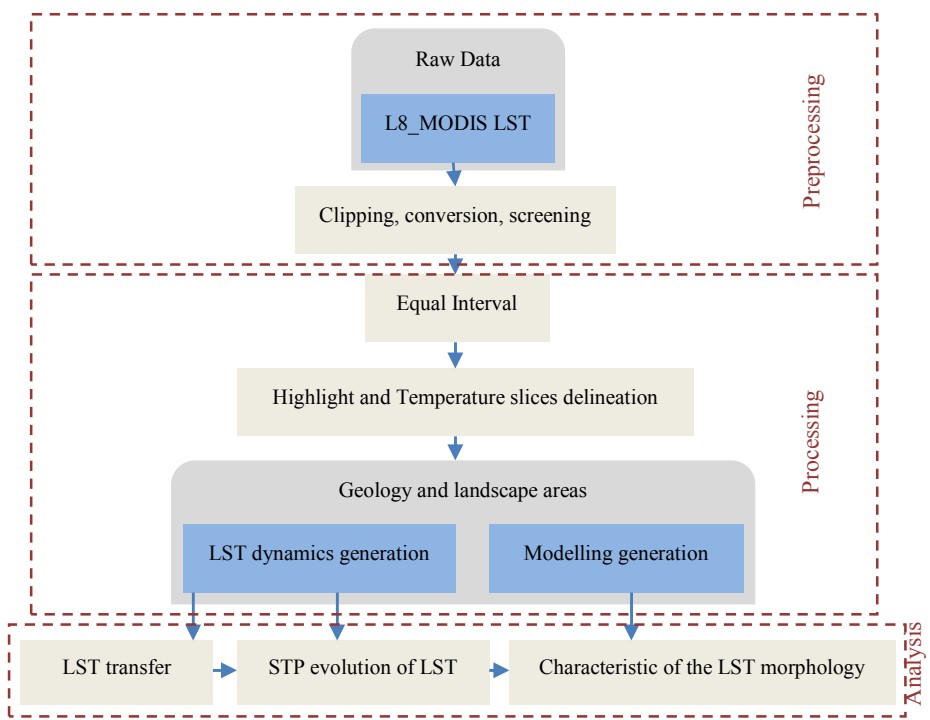

**Figure 2.** Diagram of validation process; L8: Landsat8; LST: Land Surface Temperature

**Table A. MODIS Data Land Surface Temperature Characteristics**

| SDS Name | Long Time | Number Type | Unit | Valid Slice | Fill Value | Scale factor | Add offset |
|---|---|---|---|---|---|---|---|
| LST_Day_1km | Daily daytime 1km grid Land-surface Temperature | Unite16 | Kelvin | 07500-65535 | 0 | 0.02 | 0 |

| Earth Science Data Type (ESDT) | Product Level | Nominal Data Array Dimension | Spatial resolution | Temporal resolution | Map projection |
|---|---|---|---|---|---|
| MOD11A1 | L3 | 1200 rows by 1200 columns | 1km (actual 0.928km) | daily | Integerized sinusoidal or Sinusoidal |





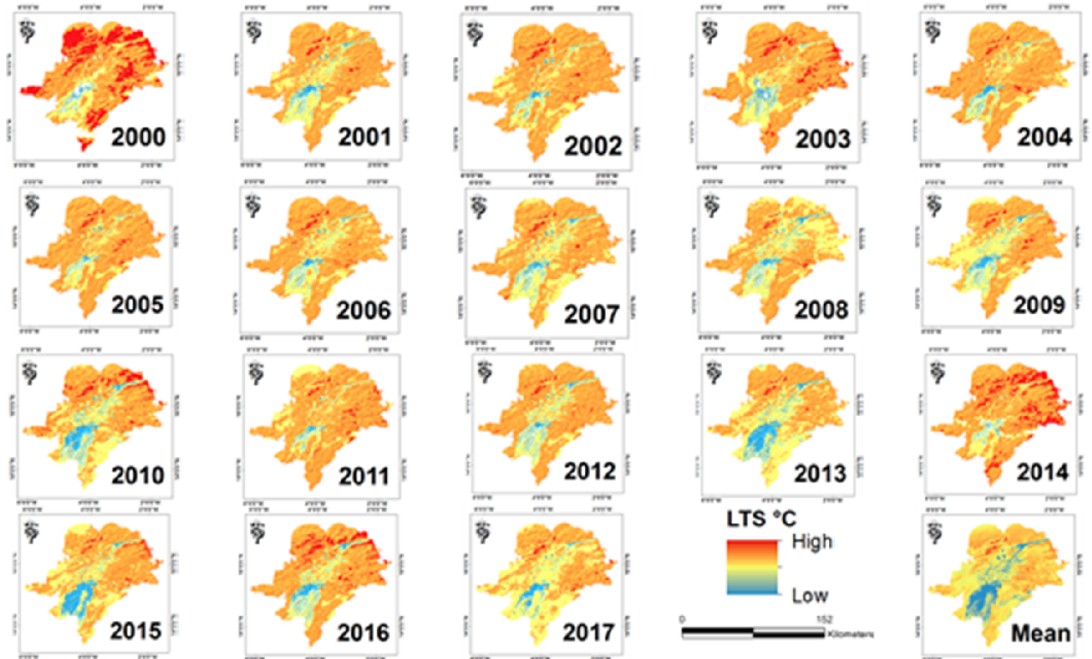

**Figure 3.** Land Surface Temperature (LST) change from 2000 to 2017

## 4 Results

### 4.1 Land surface Temperature as seen through MODIS images after processing

5   According to the general temperature change on figure 3, the LST increases from Southwest to North direction while to highlight certain characteristic places in the "Dunes-sandy, clays and laterite" areas in the Goundam zone (Dembélé & Ye, 2017; Inger et al., 2006). Some of these characteristic places located in the faguibine lake, kanango lake and Gouber lake from the subsequent works, have been already marked by a volcanic activity (El Abbass et al., 1993; Svensen et al., 2003). During these eighteen years, means of slices of maximum and minimum temperatures "43.00°C to 48.94°C and 19.21°C to 25.15°C"

10  occupy smallest surfaces, while the biggest surfaces are occupied by the mean temperature slice "31.10°C to 37.05°C". In the center of the recent alluvions has been detected the minimum mean temperature slice "19.21 ° C to 25.15 ° C" due to the existence of rivers. Temperature slices "31.10°C to 37.05°C and 25.15°C to 31.10°C" are exhibited in the major part on the recent alluvions mainly in the vegetation part toward the south of the study area.

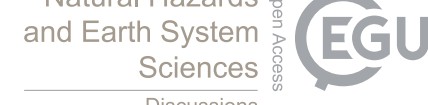



The mean of the slices of maximal temperature "43.00°C to 48.94°C" occupies 25.60% representing the larger surface in 2000 whose maximum temperature rises to 45.99 °C while the biggest surface 79.19 % in 2014 on the mean of the slices of temperature "37.05°C to 43.00°C" has the maximum temperature 49.93°C. In the overall matrix, smallest surfaces are observed in the maximum temperature slice of "43.00°C to 48.94°C" and more specifically in 2013 (0.73%). During this period (2000

5   to 2017), the year of 2013 has registered the hottest surfaces with 55.03°C as maximal temperature, while the minimal temperature 8.33°C has been register in 2000 (figure 4). From the mean statistics of surfaces, the slice "37.05°C to 43.00°C" keeps the biggest area with 64.59%, while the minimum and the maximum slices "19.21°C to 25.15°C" and "43.00°C to 48.94°C" occupy the smallest surfaces respectively with  2.02% and 4.02%, which implies an extension of desertification. Therefore, the maximum temperature increases year by year with an annual progression of 0.24°C while the surface occupied

10  by the highest temperature decreases of -1.42% by year. This means strong transitions between surfaces of the local slices of temperatures of different years, and consequently to the determination of the hottest areas located in the North part.

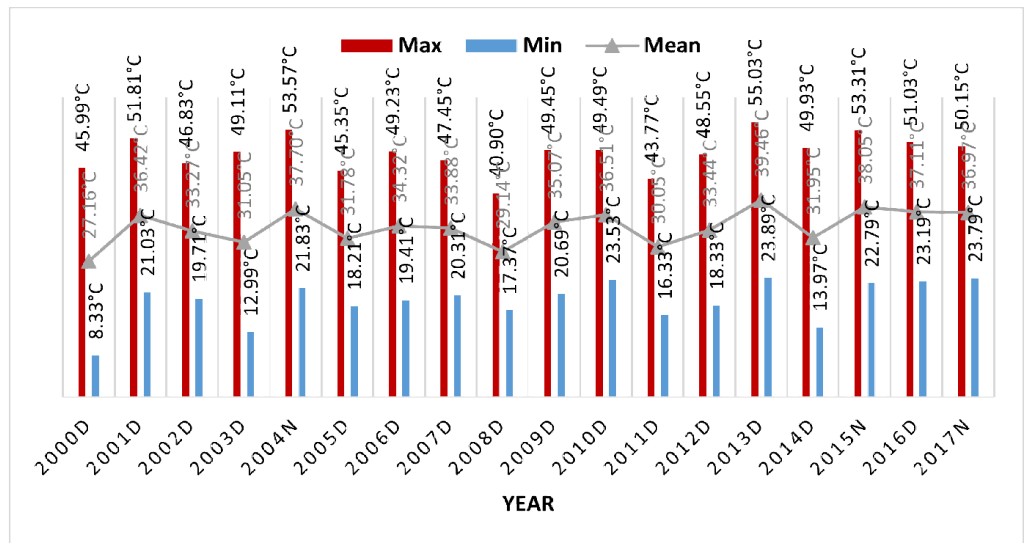

**Figure 4.** Land surface temperature over the period (2000-2017). 2000d: December 2000; 2004n: November 2004




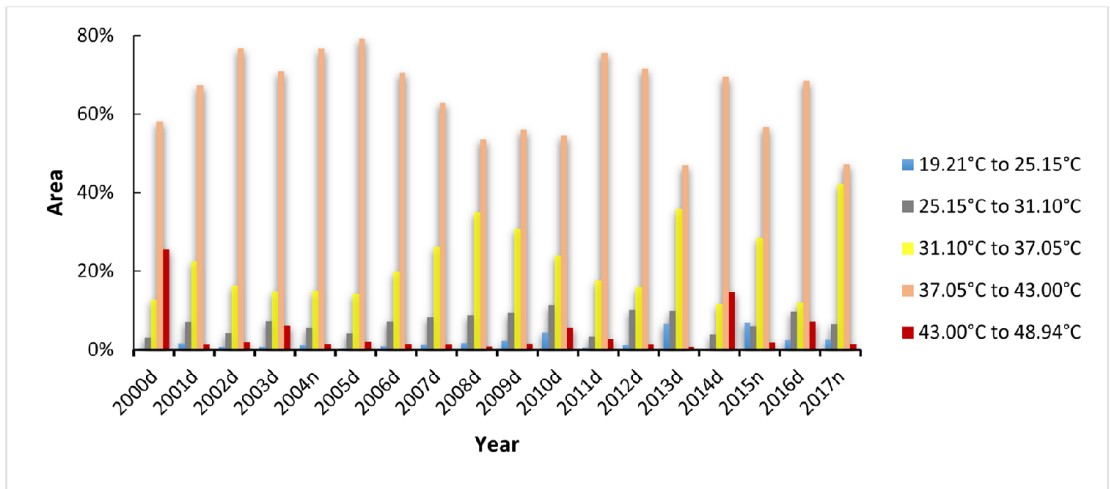

**Figure 5.** Occupancy rate of temperature slices over the period 2000 to 2017. 2000d: December 2000; 2004n: November 2004

- **The characteristic of the LST morphology**

The characteristic of the LST morphology is further builted as based on the continuous and smooth modelling. The 3D format

of the Mean of 18 years is illustrated in Figure 6. The morphology of the modelling is well described over the area. The diversified mean LTS values "from 19.21°C to 48.94°C" show from the geology and landscape configuration an appreciable demonstration. The morphological typical shapes such as concave, ridge, and flat part are illustrated in Figure 6 by taking the mean as an example. The three regions has revealed that the LST of volcanic and some mountains areas is the highest, followed by flat bare soil, and water areas (Dembélé & Ye, 2017; Mahe, Orange, Mariko, & Bricquet, 2011; Maiga. H, Marie. J, Morand.

P, N'Djim. H, 2007). Accordingly, the first region performs as a concave shape as a water area surrounded by wet area included between "19.21°C to 31.10°C" in the recent alluvions (Dembélé & Ye, 2017). Also mainly as a semi-dry area, the second region demonstrates as a flat part between "31.10°C to 43.00°C" into the arrangement of precambrian and continental terminal formations at southwest and at the East of recent alluvions. The third region reveals a ridge morphology as it is a mixed Dunes, sandy, clays, and laterite at the northwest of the study area include between "43.00°C to 48.94°C". Therefore, being an occupied

land in major part by recent alluvions area makes the 3D format of the first region presents as a concave shape, while the region in ridge shape is constituted by the volcanic hazards area.





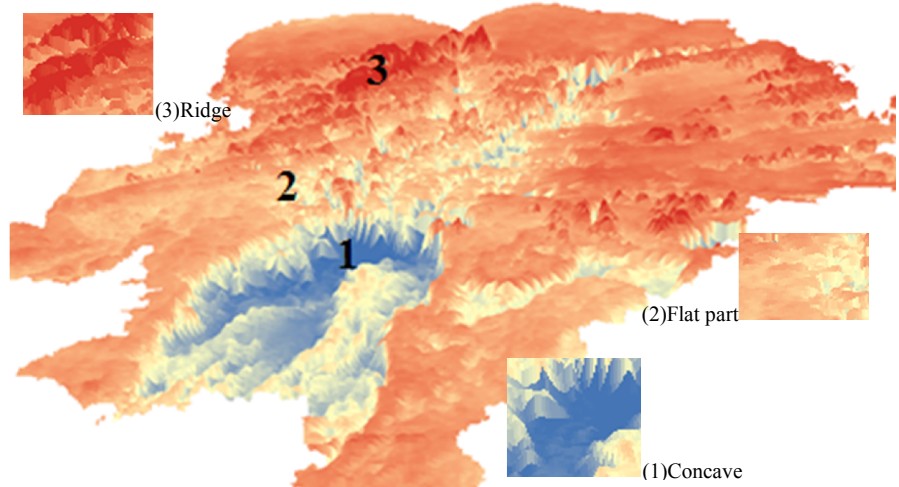

**Figure 6.** Modelling of the mean LST

## 4.2 Composition of temperature slices at the environmental scale

The evolution of LST between 2000 and 2017 generally shows a decrease in the extent of the highest temperature slices (Figure 5). Indeed, the maximum slices "37.05 ° C to 43.00 ° C and 43.00 ° C to 48.94 ° C" which, in 2000, were the dominant matrices of the environment respectively with 25.60% and 58.11% (figure 5), undergo an annual regression respectively -0.64% and -1.24% towards the north (figure 7), with a high concentration of the maximum slice "43.00 ° C to 48.94 ° C" in the Faguibine lake. On the other hand, the slice "31.10 ° C to 37.05 ° C" experienced a sharp spatial increase over the same period, from 12.72% of the total area surveyed in 2000 to 42.25% in 2017, a respective annual increase of 1.74% which made of it the dominant matrix in temperature of the environment. The total areas of the minimum "19.21 ° C to 25.15 ° C and 25.15 ° C to 31.10 ° C" slices experienced slight spatial increases of 0.44% and 3.14% respectively up to 2.57% and 6.47% over the same period (2000 to 2017), with a rate of annual increase of 0.13% and 0.20% of these respective minimum slices.





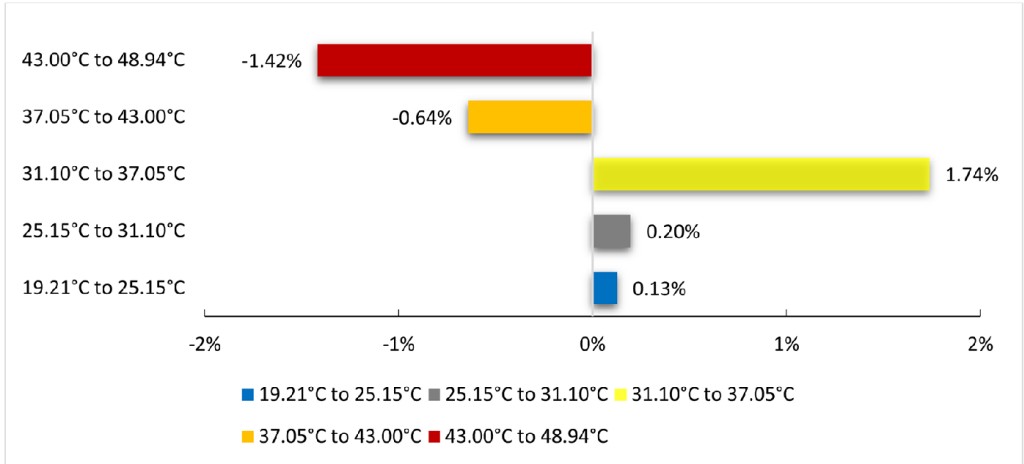

**Figure 7.** Mean annual evolution rate from 2000 to 2017

## 4.3 Land surface temperature (LST) transfer

Considering transited surfaces of the temperature slices, 42.03% of the 58.11% of the areas occupied by the fourth temperature slice "30.93 to 38.46" in 2000 remained intact in the fourth slices "39.5 to 45.65" in 2001, while 0.05% of "30.93 to 38.46" in 2000 are converted to "45.65 to 51.81" (maximum slice of 2001), then 0.34% and 15.39% were fully converted respectively to "27.19 to 33.34 and 33.34 to 39.5" in 2001 (Table B (1)). Finally, the minimum slices "8.33 to 15.86" of 2000, with a 0.28% permanence rate in the minimum slices of 2001 "21.03 to 27.19", constitute the least stable slices in the environment. Globally, this period from 2000 to 2001 is characterized by three main types of temperature dynamics: a stability of the grounds occupation  (50.60% of the space), a dynamics of narrowing of low temperature (46.37% of the space), and an opening of High temperatures (2.63 %) expressing varying degrees ("8.33 to 15.86" into "27.19 to 33.34": 0.15%; "15.86 to 23.39" into "33.34 to 39.5" and 39.5 to 45.65: respectively 0.45% and 0.02%, "23.39 to 30.93" into "39.5 to 45.65": 1.96%). These dynamics are also observed over all eighteen years from 2000-2001 to 2016-2017, but in different proportions (table B).

At the global scale of the study (from 2000 to 2017) (Table B (3)), the slice "30 .93 to 38.46" has decreased by approximately 0.16% in favor of the maximum slice "44.88 to 50.15" of 2017. Indeed, on the 58.11% of the areas occupied by the "30.93 to 38.46" slice in 2000, 26.67% remained intact with the "39.61 to 44.88" slice of 2017; 30.50% became the "34.33 to 39.61" slice and 0.53% were totally converted to "29.06 to 34.33". The proportion of the "23.39 to 30.93" portion of 2000 has increased from 12.72% to 58.08% of the slice "34.33 to 39.61", or 64.88% increase. This "34.33 to 39.61" slice is now the new matrix of an LST previously dominated by the "30.93 to 38.46" slice. The same progressive trend is observed in the slice "15.86 to 23.39" of 2000 with a rate of 7.05% at the expense of minimum slices. In fact, on the 58.08% of the area occupied by the slice "34.33 to 39.61" in 2006, 30.50% were from the "30.93 to 38.46" slice and 20.21% were from the maximum slice in 2000



"38.46 to 45.99". From the same period, 35.03% of the areas have not changed class (stability), while 1.84% of the areas have evolved in terms of reconstituting the maximum slice through a process of temperature increase, and 62.77% of areas have undergone a change in the LST slices. The increase in temperature becomes the most important phenomenon with the surface shrinkage of the maximum slices and the surface widening of the slice "39.61 to 44.88". This leads to terrestrial global

warming, and especially to the signalization of volcanic manifestation suspects into Faguibine Lake as well the degradation of the global environment.

## 4.4 Evolution status of LST and spatial structure dynamics

Between 2000 and 2001, the mean maximum temperature slice "43.00 ° C to 48.94 ° C" undergone a decrease in the number of spots in parallel with a decrease in the total area (Table C (1)). It appears obvious, based on the value of the decision tree

(Bogaert, Ceulemans, & Salvador-Van Eysenrode, 2004), that the dominant transformation process of this slice was attrition (disappearance of spots). The decrease in the number of spots is associated with a very sharp decrease in the area of the maximum slice. On the other hand, the creation of new spots is the dominant transformation process in the mean temperature slices "19.21 ° C to 25.15 ° C", "25.15 ° C to 31.10 ° C", "31.10 ° C to 37.05 ° C" and "37.05 ° C to 43.00 ° C", with an increase in the number of spots but also in the total area over this period (Tables 3). Between the years 2001 and 2002, the process of

attrition is observed in the slices "19.21 ° C to 25.15 ° C", "25.15 ° C to 31.10 ° C" and "31.10 ° C to 37.05 ° C", while the creation process has concerned the "37.05 ° C to 43.00 ° C" and "43.00 ° C to 48.94 ° C" slices, as the increase in the number of spots in 2001 was accompanied by an increase in the total area of these slices. As for the minimum slices ("19.21 ° C to 25.15 ° C" and "25.15 ° C to 31.10 ° C") and the maximum slice "43.00 ° C to 48.94 ° C", the decrease in the number of spots and the area in 2003 compared to 2002, suggests a process of spots attrition. Consequently, during all the period of 2000 to

2017, the spatial transformation processes (STP) are also observed with a sequence of creation and attrition of spots but in different proportions (Table C). In summary, dominant spatial transformation processes constitute 53.33% of spots creation (formation of new spots) followed by 46.66% of spot attrition expressed in different levels (figure 8). Thus the maximum temperature slices "43.00 ° C to 48.94 ° C" undergone more disappearance of the spots (attrition) up to 12.22%  where the shrinking of its surface until the volcanic hazards zone (Faguibine Lake).





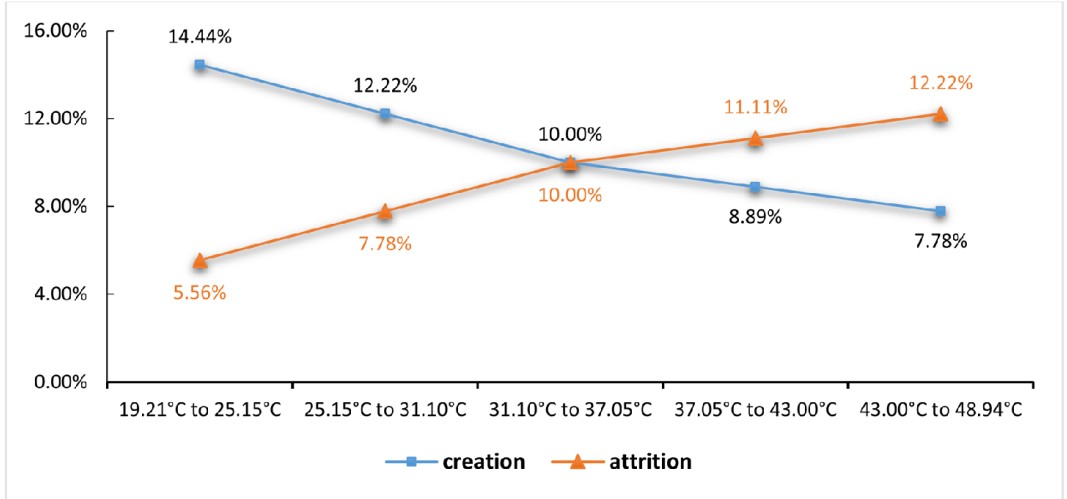

**Figure 8.** Spatial transformation processes (STP) evolution

## Conclusion

A diachronic analysis of MODIS satellite images over a period of eighteen years from 2000 to 2017 was conducted to evaluate the dynamics of Land Surface Temperature and its impact on the environment. The "equal interval" method used has permitted

to discriminate  five temperature slices occupying the study area and their mean areas occupied  respectively 2.02% ("19.21 ° C to 25.15 ° C"), 7.00% ("25.15 ° C to 31.10 ° C "), 21.97% (" 31.10 ° C to 37.05 ° C "), 64.59% (" 37.05 ° C to 43.00 ° C "), and 4.42% (" 43.00 ° C to 48.94 ° C "). The analysis of the LST carried out over the period 2000-2017 makes it possible to identify a temporal and spatial evolution of the occupation spaces of the temperature slices. The calculated mean annual evolution rates show a progression of the first three mean temperature slices "19.21 ° C to 25.15 ° C", "25.15 ° C to 31.10 °

C", and "31.10 ° C to 37.05 ° C" respectively 0.13 %, 0.20%, and 1.74%; and a regression of the mean temperature slices "37.05 ° C to 43.00 ° C", and "43.00 ° C to 48.94 ° C" respectively -0.64%, -1.42%. Similarly, a strong accentuation of the maximum slice "43.00 ° C to 48.94 ° C" in the Faguibine Lake was observed and proved by 12.22% attrition as the dominant spatial transformation process of the maximum slice. As a result, terrestrial global warming takes up a lot of space in the IND with the determination of volcanic areas, followed by the degradation of the global environment and the expansion of

desertification. Thus, this work has valuable elements for future studies of terrestrial global warming.

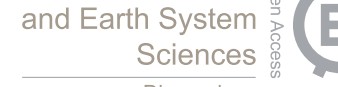



**Author Contributions**

Dembélé Abdramane conceived and designed the whole study from the framework to the elaboration of the Analysis of Land surface Temperature change using MODIS data in the Inner Delta of Niger. Touré Amadou contributed to the spatial analysis and professor Xiufen Ye revised the paper

5 **Acknowledgments**

Authors wish to thank the laboratory of biomimetic micro robot and system of Harbin Engineering University (China), and the Geodesy department of National School of Engineers of Mali (ENI-Ecole Nationale d'Ingénieurs).

**Appendix**

25

30



**Table B 1.** LST areas Transition matrix from 200 to 2017 periods

| Year | | | | 2001 | | | |
|---|---|---|---|---|---|---|---|
| | T-Slice °C | 21.03 to 27.19 | 27.19 to 33.34 | 33.34 to 39.5 | 39.5 to 45.65 | 45.65 to 51.81 | Total |
| **2000** | 8.33 to 15.86 | **0.283%** | 0.153% | 0.004% | 0.000% | 0.000% | 0.44% |
| | 15.86 to 23.39 | 0.627% | **2.030%** | 0.435% | 0.024% | 0.000% | 3.14% |
| | 23.39 to 30.93 | 0.352% | 4.159% | **6.198%** | 1.963% | 0.000% | 12.72% |
| | 30.93 to 38.46 | 0.006% | 0.337% | 15.388% | **42.083%** | 0.052% | 58.11% |
| | 38.46 to 45.99 | 0.010% | 0.420% | 23.813% | 1.262% | **0.000%** | 25.60% |
| | Total | 1.28% | 7.10% | 45.84% | 45.33% | 0.05% | 100.01% |

| Year | | | | 2002 | | | |
|---|---|---|---|---|---|---|---|
| **2001** | 21.03 to 27.19 | **0.656%** | 0.866% | 0.066% | 0.056% | 0.001% | 1.68% |
| | 27.19 to 33.34 | 0.043% | **3.270%** | 2.777% | 0.988% | 0.020% | 7.13% |
| | 33.34 to 39.5 | 0.001% | 0.124% | **11.333%** | 10.998% | 0.016% | 22.51% |
| | 39.5 to 45.65 | 0.001% | 2.041% | 64.466% | **0.803%** | 0.000% | 67.37% |
| | 45.65 to 51.81 | 0.001% | 0.222% | 1.087% | 0.000% | **0.000%** | 1.31% |
| | Total | 0.70% | 6.52% | 79.73% | 12.85% | 0.04% | 100.00% |

| Year | | | | 2003 | | | |
|---|---|---|---|---|---|---|---|
| | T-Slice °C | 12.99 to 20.21 | 20.21 to 27.44 | 27.44 to 34.66 | 34.66 to 41.89 | 41.89 to 49.11 | total |
| **2002** | 19.71 to 25.13 | **0.098%** | 0.520% | 0.014% | 0.006% | 0.002% | 0.72% |
| | 25.13 to 30.56 | 0.283% | **3.077%** | 0.700% | 0.041% | 0.001% | 4.24% |
| | 30.56 to 35.98 | 0.120% | 1.943% | **7.538%** | 6.561% | 0.003% | 16.34% |
| | 35.98 to 41.41 | 0.245% | 1.649% | 6.536% | **63.414%** | 4.401% | 76.72% |
| | 41.41 to 46.83 | 0.002% | 0.014% | 0.020% | 0.411% | **1.542%** | 1.99% |
| | Total | 0.75% | 7.20% | 14.81% | 70.43% | 5.95% | 100.01% |

| Year | | | | 2004 | | | |
|---|---|---|---|---|---|---|---|
| | T-Slice °C | 21.83 to 28.18 | 28.18 to 34.53 | 34.53 to 40.87 | 40.87 to 47.22 | 47.22 to 53.57 | Total |
| **2003** | 12.99 to 20.21 | **0.256%** | 0.180% | 0.086% | 0.239% | 0.009% | 0.77% |
| | 20.21 to 27.44 | 0.748% | **3.634%** | 1.391% | 1.476% | 0.004% | 7.26% |
| | 27.44 to 34.66 | 0.006% | 1.648% | **6.490%** | 6.682% | 0.046% | 14.89% |
| | 34.66 to 41.89 | 0.002% | 0.065% | 6.844% | **63.407%** | 0.550% | 70.95% |
| | 41.89 to 49.11 | 0.000% | 0.003% | 0.078% | 5.164% | **0.871%** | 6.13% |
| | total | 1.01% | 5.53% | 14.89% | 76.97% | 1.48% | 100.00% |

| Year | | | | 2005 | | | |
|---|---|---|---|---|---|---|---|
| | T-Slice °C | 18.21 to 23.64 | 23.64 to 29.07 | 29.07 to 34.49 | 34.49 to 39.92 | 39.92 to 45.35 | Total |
| **2004** | 21.83 to 28.18 | **0.351%** | 0.779% | 0.002% | 0.000% | 0.001% | 1.14% |
| | 28.18 to 34.53 | 0.042% | **2.847%** | 2.678% | 0.062% | 0.007% | 5.65% |
| | 34.53 to 40.87 | 0.023% | 0.337% | **9.214%** | 5.465% | 0.004% | 15.08% |
| | 40.87 to 47.22 | 0.008% | 0.076% | 2.317% | **73.096%** | 1.057% | 76.62% |
| | 47.22 to 53.57 | 0.008% | 0.567% | 0.935% | 0.000% | **0.000%** | 1.51% |
| | Total | 0.43% | 4.61% | 15.15% | 78.62% | 1.07% | 100.00% |

| Year | | | | 2006 | | | |
|---|---|---|---|---|---|---|---|
| | T-Slice °C | 19.41 to 25.37 | 25.37 to 31.34 | 31.34 to 37.3 | 37.3 to 43.27 | 43.27 to 49.23 | Total |
| **2005** | 18.21 to 23.64 | **0.379%** | 0.038% | 0.003% | 0.000% | 0.000% | 0.43% |
| | 23.64 to 29.07 | 0.499% | **3.455%** | 0.094% | 0.001% | 0.000% | 4.06% |
| | 29.07 to 34.49 | 0.007% | 3.572% | **9.644%** | 0.985% | 0.000% | 14.23% |
| | 34.49 to 39.92 | 0.001% | 0.148% | 10.088% | **68.507%** | 0.383% | 79.19% |
| | 39.92 to 45.35 | 0.002% | 0.007% | 0.024% | 1.028% | **1.013%** | 2.08% |
| | Total | 0.89% | 7.22% | 19.85% | 70.52% | 1.40% | 99.99% |

Each value of the table corresponds to a fraction of the converted Land Surface Temperature (LST) area, between year and the following (example= 2000 and 2001), of the Temperature Slice (T-Slice) indicated on the line to the T-Slice at the head of the column. For example, 0.153% expresses the fraction of the LST area belonging to the "8.33 to 15.86" slice in 2000 and which was converted to the "27.19 to 33.34" slice in 2001. Bold values indicate class permanence. Above the diagonal are the dynamics of hottest surfaces.



**Table B 2.** LST areas Transition matrix from 200 to 2017 periods

| Year | | | 2007 | | | | |
|---|---|---|---|---|---|---|---|
| | T-Slice °C | 20.31 to 25.74 | 25.74 to 31.17 | 31.17 to 36.59 | 36.59 to 42.02 | 42.02 to 47.45 | Total |
| 2006 | 19.41 to 25.37 | **0.721%** | 0.178% | 0.000% | 0.000% | 0.000% | 0.91% |
| | 25.37 to 31.34 | 0.537% | **6.144%** | 0.517% | 0.005% | 0.000% | 7.24% |
| | 31.34 to 37.3 | 0.002% | 1.971% | **15.038%** | 2.839% | 0.000% | 19.87% |
| | 37.3 to 43.27 | 0.002% | 0.057% | 10.513% | **59.549%** | 0.377% | 70.58% |
| | 43.27 to 49.23 | 0.003% | 0.494% | 0.912% | 0.000% | **0.000%** | 1.41% |
| | Total | 1.27% | 8.84% | 26.98% | 62.39% | 0.38% | 100.01% |

| Year | | | 2008 | | | | |
|---|---|---|---|---|---|---|---|
| | T-Slice °C | 17.37 to 22.07 | 22.07 to 26.78 | 26.78 to 31.48 | 31.48 to 36.19 | 36.19 to 40.9 | Total |
| 2007 | 20.31 to 25.74 | **0.959%** | 0.266% | 0.002% | 0.002% | 0.000% | 1.28% |
| | 25.74 to 31.17 | 0.662% | **6.040%** | 1.378% | 0.022% | 0.000% | 8.37% |
| | 31.17 to 36.59 | 0.034% | 2.071% | **16.176%** | 7.703% | 0.000% | 26.14% |
| | 36.59 to 42.02 | 0.022% | 0.316% | 17.279% | **44.742%** | 0.341% | 62.90% |
| | 42.02 to 47.45 | 0.001% | 0.012% | 0.023% | 0.764% | **0.495%** | 1.30% |
| | Total | 1.68% | 8.70% | 34.86% | 53.23% | 0.84% | 99.99% |

| Year | | | 2009 | | | | |
|---|---|---|---|---|---|---|---|
| | T-Slice °C | 20.69 to 26.44 | 26.44 to 32.19 | 32.19 to 37.95 | 37.95 to 43.7 | 43.7 to 49.45 | Total |
| 2008 | 17.37 to 22.07 | **1.275%** | 0.368% | 0.013% | 0.008% | 0.003% | 1.72% |
| | 22.07 to 26.78 | 0.993% | **6.526%** | 0.926% | 0.252% | 0.013% | 8.79% |
| | 26.78 to 31.48 | 0.002% | 2.190% | **16.643%** | 16.116% | 0.024% | 35.02% |
| | 31.48 to 36.19 | 0.011% | 13.034% | 39.480% | **0.983%** | 0.000% | 53.60% |
| | 36.19 to 40.9 | 0.000% | 0.355% | 0.514% | 0.000% | **0.000%** | 0.87% |
| | total | 2.28% | 22.47% | 57.58% | 17.36% | 0.04% | 100.00% |

| Year | | | 2010 | | | | |
|---|---|---|---|---|---|---|---|
| | T-Slice °C | 23.53 to 28.72 | 28.72 to 33.91 | 33.91 to 39.11 | 39.11 to 44.3 | 44.3 to 49.49 | Total |
| 2009 | 20.69 to 26.44 | **2.106%** | 0.187% | 0.010% | 0.000% | 0.000% | 2.31% |
| | 26.44 to 32.19 | 2.251% | **6.508%** | 0.533% | 0.016% | 0.000% | 9.31% |
| | 32.19 to 37.95 | 0.010% | 4.582% | **15.530%** | 10.559% | 0.103% | 30.84% |
| | 37.95 to 43.7 | 0.048% | 7.876% | 43.482% | **4.512%** | 0.000% | 55.98% |
| | 43.7 to 49.45 | 0.011% | 0.026% | 0.590% | 0.931% | **0.000%** | 1.56% |
| | Total | 4.43% | 19.18% | 60.14% | 16.02% | 0.10% | 100.00% |

| Year | | | 2011 | | | | |
|---|---|---|---|---|---|---|---|
| | T-Slice °C | 16.33 to 21.82 | 21.82 to 27.31 | 27.31 to 32.79 | 32.79 to 38.28 | 38.28 to 43.77 | Total |
| 2010 | 23.53 to 28.72 | **0.539%** | 2.446% | 0.899% | 0.476% | 0.000% | 4.45% |
| | 28.72 to 33.91 | 0.018% | **0.894%** | 4.940% | 5.347% | 0.056% | 11.35% |
| | 33.91 to 39.11 | 0.000% | 0.037% | **5.864%** | 17.966% | 0.114% | 24.03% |
| | 39.11 to 44.3 | 0.000% | 5.624% | 47.460% | **1.398%** | 0.000% | 54.55% |
| | 44.3 to 49.49 | 0.076% | 4.482% | 1.062% | 0.000% | **0.000%** | 5.63% |
| | Total | 0.63% | 13.48% | 60.23% | 25.19% | 0.17% | 100.01% |

| Year | | | 2012 | | | | |
|---|---|---|---|---|---|---|---|
| | T-Slice °C | 18.33 to 24.37 | 24.37 to 30.42 | 30.42 to 36.46 | 36.46 to 42.51 | 42.51 to 48.55 | total |
| 2011 | 16.33 to 21.82 | **0.401%** | 0.147% | 0.000% | 0.000% | 0.000% | 0.56% |
| | 21.82 to 27.31 | 0.493% | **2.854%** | 0.033% | 0.000% | 0.000% | 3.40% |
| | 27.31 to 32.79 | 0.130% | 4.220% | **5.495%** | 7.667% | 0.000% | 17.63% |
| | 32.79 to 38.28 | 0.053% | 2.766% | 10.230% | **61.759%** | 0.599% | 75.65% |
| | 38.28 to 43.77 | 0.002% | 0.034% | 0.057% | 1.946% | **0.717%** | 2.76% |
| | Total | 1.08% | 10.02% | 15.81% | 71.37% | 1.32% | 100.00% |





**Table B 3.** LST areas Transition matrix from 200 to 2017 periods

| Year | | | | 2013 | | | |
|---|---|---|---|---|---|---|---|
| | T-Slice °C | 23.89 to 30.12 | 30.12 to 36.35 | 36.35 to 42.57 | 42.57 to 48.8 | 48.8 to 55.03 | Total |
| 2012 | 18.33 to 24.37 | **1.028%** | 0.080% | 0.021% | 0.008% | 0.000% | 1.17% |
| | 24.37 to 30.42 | 5.272% | **4.265%** | 0.482% | 0.043% | 0.003% | 10.11% |
| | 30.42 to 36.46 | 0.080% | 5.266% | **9.734%** | 0.756% | 0.004% | 15.89% |
| | 36.46 to 42.51 | 0.003% | 0.244% | 25.624% | **45.361%** | 0.154% | 71.48% |
| | 42.51 to 48.55 | 0.026% | 0.748% | 0.586% | 0.000% | **0.000%** | 1.36% |
| | total | 6.41% | 10.60% | 36.45% | 46.17% | 0.16% | 100.01% |

| Year | | | | 2014 | | | |
|---|---|---|---|---|---|---|---|
| | T-Slice °C | 13.97 to 21.16 | 21.16 to 28.35 | 28.35 to 35.55 | 35.55 to 42.74 | 42.74 to 49.93 | Total |
| 2013 | 23.89 to 30.12 | **0.183%** | 3.124% | 2.783% | 0.295% | 0.001% | 6.54% |
| | 30.12 to 36.35 | 0.010% | **0.340%** | 5.707% | 3.538% | 0.015% | 9.90% |
| | 36.35 to 42.57 | 0.000% | 0.231% | **2.266%** | 31.823% | 1.028% | 35.91% |
| | 42.57 to 48.8 | 0.068% | 0.713% | 33.158% | **12.714%** | 0.000% | 46.91% |
| | 48.8 to 55.03 | 0.086% | 0.644% | 0.000% | 0.000% | **0.000%** | 0.73% |
| | Total | 0.35% | 5.05% | 43.92% | 48.37% | 1.04% | 99.99% |

| Year | | | | 2015 | | | |
|---|---|---|---|---|---|---|---|
| | T-Slice °C | 22.79 to 28.89 | 28.89 to 35 | 35 to 41.1 | 41.1 to 47.21 | 47.21 to 53.31 | Total |
| 2014 | 13.97 to 21.16 | **0.184%** | 0.024% | 0.001% | 0.000% | 0.000% | 0.21% |
| | 21.16 to 28.35 | 3.050% | **0.363%** | 0.318% | 0.132% | 0.000% | 3.87% |
| | 28.35 to 35.55 | 3.692% | 3.697% | **2.859%** | 1.343% | 0.015% | 11.62% |
| | 35.55 to 42.74 | 0.055% | 1.631% | 24.758% | **42.842%** | 0.182% | 69.57% |
| | 42.74 to 49.93 | 0.010% | 0.029% | 0.183% | 12.792% | **1.685%** | 14.73% |
| | Total | 6.99% | 5.74% | 28.12% | 57.11% | 1.88% | 100.00% |

| Year | | | | 2016 | | | |
|---|---|---|---|---|---|---|---|
| | T-Slice °C | 23.19 to 28.76 | 28.76 to 34.33 | 34.33 to 39.89 | 39.89 to 45.46 | 45.46 to 51.03 | total |
| 2015 | 22.79 to 28.89 | **2.065%** | 4.655% | 0.267% | 0.010% | 0.000% | 7.00% |
| | 28.89 to 35 | 0.197% | **3.513%** | 2.186% | 0.036% | 0.002% | 5.94% |
| | 35 to 41.1 | 0.110% | 1.258% | **7.942%** | 18.987% | 0.068% | 28.42% |
| | 41.1 to 47.21 | 0.076% | 0.330% | 1.715% | **48.918%** | 5.598% | 56.73% |
| | 47.21 to 53.31 | 0.003% | 0.003% | 0.484% | 1.415% | **0.000%** | 1.92% |
| | total | 2.45% | 9.76% | 12.59% | 69.37% | 5.67% | 100.01% |

| Year | | | | 2017 | | | |
|---|---|---|---|---|---|---|---|
| | T-Slice °C | 23.79 to 29.06 | 29.06 to 34.33 | 34.33 to 39.61 | 39.61 to 44.88 | 44.88 to 50.15 | Total |
| 2016 | 23.19 to 28.76 | **1.650%** | 0.468% | 0.259% | 0.091% | 0.000% | 2.47% |
| | 28.76 to 34.33 | 0.875% | **5.020%** | 3.516% | 0.336% | 0.000% | 9.75% |
| | 34.33 to 39.89 | 0.015% | 0.896% | **10.354%** | 0.812% | 0.000% | 12.09% |
| | 39.89 to 45.46 | 0.011% | 0.030% | 28.061% | **40.136%** | 0.169% | 68.49% |
| | 45.46 to 51.03 | 0.046% | 5.842% | 1.813% | 1.292% | **0.000%** | 7.19% |
| | Total | 2.60% | 12.26% | 44.00% | 42.67% | 0.17% | 99.99% |

| Year | | | | 2017 | | | |
|---|---|---|---|---|---|---|---|
| | T-Slice °C | 23.79 to 29.06 | 29.06 to 34.33 | 34.33 to 39.61 | 39.61 to 44.88 | 44.88 to 50.15 | Total |
| 2000 | 8.33 to 15.86 | **0.387%** | 0.047% | 0.006% | 0.000% | 0.000% | 0.44% |
| | 15.86 to 23.39 | 1.355% | **1.158%** | 0.553% | 0.064% | 0.000% | 3.14% |
| | 23.39 to 30.93 | 0.391% | 4.491% | **6.804%** | 1.001% | 0.009% | 12.72% |
| | 30.93 to 38.46 | 0.001% | 0.525% | 30.501% | **26.685%** | 0.159% | 58.11% |
| | 38.46 to 45.99 | 0.006% | 3.970% | 20.217% | 1.307% | **0.000%** | 25.60% |
| | total | 2.14% | 10.19% | 58.08% | 29.06% | 0.17% | 100.01% |





**Table C 1**. Spatial Transformation Process (STP) of the LST areas from 2000-2017

| Year | LST-Mean-Slice | $n_2 - n_1$ | $a_2 - a_1$ | $P_2 - P_1$ | $a_2 / a_1$ | STP state |
|---|---|---|---|---|---|---|
| **2001-2000** | 19.21°C to 25.15°C | 1500 | 1635.45 | 44327.81 | 3.82 | creation |
| | 25.15°C to 31.10°C | 4895 | 5262.69 | 204850.30 | 2.27 | creation |
| | 31.10°C to 37.05°C | 12140 | 12913.38 | 473701.63 | 1.77 | creation |
| | 37.05°C to 43.00°C | 12776 | 12220.36 | -176579.67 | 1.16 | creation |
| | 43.00°C to 48.94°C | -28516 | -32031.88 | -546300.06 | 0.05 | attrition |
| **Year** | | | | | | |
| **2002-2001** | 19.21°C to 25.15°C | -1150 | -1266.21 | -38533.41 | 0.43 | attrition |
| | 25.15°C to 31.10°C | -3486 | -3812.08 | -118236.16 | 0.59 | attrition |
| | 31.10°C to 37.05°C | -7434 | -8139.56 | -44749.38 | 0.73 | attrition |
| | 37.05°C to 43.00°C | 11220 | 12321.28 | 135570.00 | 1.14 | creation |
| | 43.00°C to 48.94°C | 812 | 896.57 | 65948.96 | 1.52 | creation |
| **Year** | | | | | | |
| **2003-2002** | 19.21°C to 25.15°C | 49 | 66.04 | 14457.14 | 1.07 | creation |
| | 25.15°C to 31.10°C | 3579 | 3983.53 | 40360.65 | 1.71 | creation |
| | 31.10°C to 37.05°C | -1869 | -1910.20 | -218598.65 | 0.91 | attrition |
| | 37.05°C to 43.00°C | -7537 | -7599.74 | -46617.17 | 0.92 | attrition |
| | 43.00°C to 48.94°C | 4935 | 5460.37 | 210398.03 | 3.08 | creation |
| **Year** | | | | | | |
| **2004-2003** | 19.21°C to 25.15°C | 455 | 487.98 | 961.76 | 1.48 | creation |
| | 25.15°C to 31.10°C | -1878 | -2123.37 | 20460.39 | 0.78 | attrition |
| | 31.10°C to 37.05°C | 369 | 250.58 | 232682.89 | 1.01 | creation |
| | 37.05°C to 43.00°C | 7492 | 7477.97 | -15434.03 | 1.08 | creation |
| | 43.00°C to 48.94°C | -5505 | -6093.16 | -238671.01 | 0.25 | attrition |
| **Year** | | | | | | |
| **2005-2004** | 19.21°C to 25.15°C | -849 | -936.34 | -26601.70 | 0.38 | attrition |
| | 25.15°C to 31.10°C | -1913 | -2096.46 | -44618.07 | 0.72 | attrition |
| | 31.10°C to 37.05°C | -1026 | -1119.16 | -44745.63 | 0.94 | attrition |
| | 37.05°C to 43.00°C | 3064 | 3399.93 | 60982.87 | 1.03 | creation |
| | 43.00°C to 48.94°C | 686 | 752.03 | 54982.53 | 1.38 | creation |
| **Year** | | | | | | |
| **2006-2005** | 19.21°C to 25.15°C | 571 | 632.88 | 15063.87 | 2.12 | creation |
| | 25.15°C to 31.10°C | 3827 | 4192.50 | 73320.59 | 1.78 | creation |
| | 31.10°C to 37.05°C | 6792 | 7433.91 | 62891.28 | 1.40 | creation |
| | 37.05°C to 43.00°C | -10369 | -11375.19 | -84196.94 | 0.89 | attrition |
| | 43.00°C to 48.94°C | -817 | -884.10 | -67078.80 | 0.68 | attrition |
| **Year** | | | | | | |
| **2007-2006** | 19.21°C to 25.15°C | 452 | 488.27 | 20028.17 | 1.41 | creation |
| | 25.15°C to 31.10°C | 1355 | 1492.38 | 22028.36 | 1.16 | creation |
| | 31.10°C to 37.05°C | 7540 | 8275.36 | 27750.79 | 1.32 | creation |
| | 37.05°C to 43.00°C | -9260 | -10111.29 | -52103.44 | 0.89 | attrition |
| | 43.00°C to 48.94°C | -128 | -144.72 | -17703.88 | 0.92 | attrition |

The Spatial Transformation Process has been calculated and its evolution status is determined by the decision tree proposed by (Bogaert, and al.2004). Therefore, knowing *n* (number of spots) *a* (area) and *p* (perimeter), the difference is applied between the year and the previous (example= 2001 and 2000) to determine the STP state. LST-Mean-Slice = Land Surface Temperature (LST) mean Slice





**Table C 2.** Spatial Transformation Process (STP) of the LST areas from 2000-2017

| Year | LST-Mean-Slice | $n_2 - n_1$ | $a_2 - a_1$ | $P_2 - P_1$ | $a_2 / a_1$ | STP state |
|---|---|---|---|---|---|---|
| **2008-2007** | 19.21°C to 25.15°C | 515 | 582.09 | 8974.75 | 1.34 | creation |
| | 25.15°C to 31.10°C | 450 | 547.79 | 33265.40 | 1.05 | creation |
| | 31.10°C to 37.05°C | 10465 | 11711.78 | 18157.52 | 1.34 | creation |
| | 37.05°C to 43.00°C | -11534 | -12272.64 | -41916.45 | 0.85 | attrition |
| | 43.00°C to 48.94°C | -524 | -569.03 | -18481.23 | 0.67 | attrition |
| Year | | | | | | |
| **2009-2008** | 19.21°C to 25.15°C | 717 | 776.17 | 17921.70 | 1.34 | creation |
| | 25.15°C to 31.10°C | 669 | 690.84 | 28703.16 | 1.06 | creation |
| | 31.10°C to 37.05°C | -4863 | -5516.56 | -33614.59 | 0.88 | attrition |
| | 37.05°C to 43.00°C | 3105 | 3137.79 | -37227.44 | 1.04 | creation |
| | 43.00°C to 48.94°C | 834 | 911.76 | 24217.16 | 1.80 | creation |
| Year | | | | | | |
| **2010-2009** | 19.21°C to 25.15°C | 2580 | 2821.79 | 12820.51 | 1.93 | creation |
| | 25.15°C to 31.10°C | 2475 | 2688.99 | -20264.99 | 1.22 | creation |
| | 31.10°C to 37.05°C | -8135 | -8984.65 | -136105.28 | 0.78 | attrition |
| | 37.05°C to 43.00°C | -1588 | -1893.17 | 21893.94 | 0.97 | attrition |
| | 43.00°C to 48.94°C | 4901 | 5367.04 | 121655.83 | 3.61 | creation |
| Year | | | | | | |
| **2011-2010** | 19.21°C to 25.15°C | -4679 | -5129.80 | -67990.11 | 0.13 | attrition |
| | 25.15°C to 31.10°C | -9581 | -10483.49 | -187719.42 | 0.30 | attrition |
| | 31.10°C to 37.05°C | -7744 | -8437.57 | 53269.70 | 0.73 | attrition |
| | 37.05°C to 43.00°C | 25235 | 27835.27 | 213736.73 | 1.39 | creation |
| | 43.00°C to 48.94°C | -3454 | -3784.40 | -11296.90 | 0.49 | attrition |
| Year | | | | | | |
| **2012-2011** | 19.21°C to 25.15°C | 721 | 804.35 | 34475.79 | 2.09 | creation |
| | 25.15°C to 31.10°C | 8054 | 8848.26 | 255001.58 | 2.97 | creation |
| | 31.10°C to 37.05°C | -2118 | -2296.92 | 49775.50 | 0.90 | attrition |
| | 37.05°C to 43.00°C | -5130 | -5509.10 | -241275.13 | 0.94 | attrition |
| | 43.00°C to 48.94°C | -1691 | -1846.59 | -97977.73 | 0.49 | attrition |
| Year | | | | | | |
| **2013-2012** | 19.21°C to 25.15°C | 6465 | 7083.33 | 83946.75 | 5.59 | creation |
| | 25.15°C to 31.10°C | -224 | -274.32 | -17825.37 | 0.98 | attrition |
| | 31.10°C to 37.05°C | 24121 | 26410.53 | 25690.55 | 2.26 | creation |
| | 37.05°C to 43.00°C | -29366 | -32388.93 | -36272.55 | 0.66 | attrition |
| | 43.00°C to 48.94°C | -746 | -830.61 | -55539.38 | 0.54 | attrition |
| Year | | | | | | |
| **2014-2013** | 19.21°C to 25.15°C | -7620 | -8349.29 | -131721.11 | 0.03 | attrition |
| | 25.15°C to 31.10°C | -7310 | -7954.07 | -268147.22 | 0.39 | attrition |
| | 31.10°C to 37.05°C | -29357 | -32039.99 | -416945.18 | 0.32 | attrition |
| | 37.05°C to 43.00°C | 26385 | 29879.32 | 299305.00 | 1.48 | creation |
| | 43.00°C to 48.94°C | 16653 | 18464.03 | 517508.50 | 20.18 | creation |



**Table C 3.** Spatial Transformation Process (STP) of the LST areas from 2000-2017

| Year | LST-Mean-Slice | $n_2 - n_1$ | $a_2 - a_1$ | $P_2 - P_1$ | $a_2 / a_1$ | STP state |
|---|---|---|---|---|---|---|
| **2015-2014** | 19.21°C to 25.15°C | 8185 | 8954.18 | 100142.90 | 33.33 | creation |
| | 25.15°C to 31.10°C | 2547 | 2729.27 | 175158.77 | 1.53 | creation |
| | 31.10°C to 37.05°C | 20389 | 22153.20 | 382319.50 | 2.45 | creation |
| | 37.05°C to 43.00°C | -14496 | -16941.72 | -198736.03 | 0.82 | attrition |
| | 43.00°C to 48.94°C | -15229 | -16894.93 | -458885.14 | 0.13 | attrition |
| **Year** | | | | | | |
| **2016-2015** | 19.21°C to 25.15°C | -5454 | -5973.21 | -38828.74 | 0.35 | attrition |
| | 25.15°C to 31.10°C | 4590 | 5026.95 | 55761.89 | 1.64 | creation |
| | 31.10°C to 37.05°C | -19665 | -21531.74 | -227287.17 | 0.43 | attrition |
| | 37.05°C to 43.00°C | 14137 | 15526.38 | -13065.89 | 1.21 | creation |
| | 43.00°C to 48.94°C | 6346 | 6951.63 | 223419.92 | 3.75 | creation |
| **Year** | | | | | | |
| **2017-2016** | 19.21°C to 25.15°C | 118 | 131.22 | -15926.67 | 1.04 | creation |
| | 25.15°C to 31.10°C | -3950 | -4328.02 | -91867.28 | 0.66 | attrition |
| | 31.10°C to 37.05°C | 36323 | 39769.84 | 288187.47 | 3.49 | creation |
| | 37.05°C to 43.00°C | -25598 | -28080.73 | 42564.71 | 0.69 | attrition |
| | 43.00°C to 48.94°C | -6841 | -7492.31 | -222958.22 | 0.21 | attrition |
| **Year** | | | | | | |
| **2017-2000** | 19.21°C to 25.15°C | 2576 | 2808.90 | 33519.42 | 5.84 | creation |
| | 25.15°C to 31.10°C | 4099 | 4391.38 | 160232.57 | 2.06 | creation |
| | 31.10°C to 37.05°C | 35928 | 38942.22 | 492380.95 | 3.32 | creation |
| | 37.05°C to 43.00°C | -11464 | -14374.20 | -169371.51 | 0.81 | attrition |
| | 43.00°C to 48.94°C | -28284 | -31768.31 | -516761.43 | 0.06 | attrition |

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
