# Peer review of "Analysis of Land surface Temperature change based on MODIS data, Case study: Inner Delta of Niger"

_Natural Hazards and Earth System Sciences, 2018_

## Referee Comment (RC1) · Anonymous Referee #1 · 18 Oct 2018

This paper analyses spatial and temporal patterns of LST over the Nile region using the MODIS LST product. In my opinion this article is not suitable for publication for several reasons:

1) The English is very poor. There are a lot of sentences that are not correctly constructed. There are a lot of words that are misused and whose meaning does not fit with the sentence where they are used. This makes reading the paper very difficult and it is very hard to fully understand the author's descriptions, discussion and conclusions. The authors should ask for a language review if they struggle with the English.

2) Although is it clear that the paper targets LST evolution over time, the purpose and

relevance of the paper does not seem very clear. For instance, the authors keep referring to volcanic activity although the link between their work and the volcanic activity topic is not clear. In the conclusions they state that LST "determine volcanic areas": this does not seem correct. Did the authors mean to say that LST values are influenced by volcanic activity, in which case this is simply an English problem?

3) The data used are not properly described. The authors refer to Landsat8 several times although they seem to be using MODIS data. I suggest that the authors should read some papers where the MODIS data is used so they batter understand how to describe the data.

4) The methods are also not clearly described. For instance, why did the authors chose those 5 classes of temperature? Why do they think they are appropriate to describe the region? Are they related to vegetation distribution or climate regions? Please clearly state your motives. I did not understand the objective and conclusions of the section "The characteristic of the LST morphology". What did the authors intended to show here?

5) The figures are not properly described. Colorbars should have actual values and not "high" and "low" (Fig. 3). Some figures don't have colorbars (Fig. 6).

6) The Conclusions section is very short. The authors should better discuss the relevance of their work to the scientific community.

7) Some references are not correct and are not correctly cited in the text. Please review your references (for instance, lines 29-30, but I found several problems throughout the text).

---

## Referee Comment (RC2) · Anonymous Referee #2 · 14 Nov 2018

In this manuscript, the authors present an analysis on the temporal dataset of MODIS satellite imagery with a particular emphasis on "Land Surface Temperature (LST)". Although the authors provide several results showing changes in the LST over the time period of 2000 to 2017, the work does not seem to meet the criteria to be published in Natural Hazards and Earth System Sciences. One of the major issues is the ambiguity of the objective of this study: although the authors describe their aim is to analyze the temporal changes in LST as explicitly shown in the manuscript title, it is unclear why the surface temperature is examined and what are the factors affecting the temperature changes. The authors only present the results from the LST or MODIS data, without proposing any other data sources such as land cover, lithology, landforms, and climate.

[Figure]

No figure regarding these potential factors is presented, while the authors describe in the text that there are some correlations with water bodies or volcanic activities. These descriptions are quite vague and hardly be validated, which is not the science. The reviewer should say that this research must be redesigned in a scientific context by adding further geospatial analyses using different datasets (land cover, meteorological observations, etc.). Also, the authors should aim the assessments of disasters (in this case, drought or volcanic activities?) rather than only showing the LST changes. Otherwise, the work does not fit the scope of this scientific journal. There are also more issues to be solved: e.g., English grammatical errors, terminologies (what are morphology? modelling? evolution?), incomplete citation formats, figure components (legends, captions, etc.), analysis methods (for instance, the "equal interval method" is not a novel analysis but just a quite general, basic classification method).

---

## Author Comment (AC1) · 14 Nov 2018

The authors thank the reviewer for thoroughly reviewing our manuscript, providing valuable suggestions to improve the manuscript. Therefore a language review has been made to the entire manuscript after all corrections addressed in the following paragraphs

The authors appreciate the comments of the critics and agree that this study focuses on the influence of the characteristic places on the variation of LST during 18 years. Accordingly, a common response has been prepared for this part, and the same is given below.

The MODIS image is a good indicator of Land surface temperature (LST) on the interface analysis in order to characterize areas. The latest decades has registered the attraction of much attention through the LST at large-scale (Roy et al., 2014; Vlassova, Pérez-Cabello, Mimbrero, Llovería, & García-Martín, 2014; Wang, Liang, & Meyers, 2008; Zorer et al., 2013) and the advancement of earth and environmental sciences in the desertification monitoring so as in the monitoring of LST evolution (Hillger & Clark, 2002a, 2002b). One of the major driving forces causing many extremes of soil anomalies whose in recent years that many regions have undergone is the terrestrial global warming (C et al., 2013; Committee, 2015; Tol, 2009). Based on meteorological stations data, many studies have quantified the global temperature of the earth (Coumou, Robinson, & Rahmstorf, 2012; Frangou, Ladle, Malhado, & Whittaker, 2010; Rahmstorf & Coumou, 2011). Then several phenomena and climate extreme have been recorded during 2011–2015 and the probability increased by a factor of ten or more, in the case of some extreme high temperatures (World Meteorological Organization, 2016a)(World Meteorological Organization, 2016b). Only temperature anomalies have concerned most analysis(Report, 2018; Weber et al., 2014). The Inner Delta of Niger (IDN), in central of Mali, through desert regions represents one of the hot areas of the African plate (Friese, 2010; Mao, K et al., 2017). Inner Delta of Niger, is part of a large geological structure (Continental Terminal, Precambrian, Quaternary) that have played several roles since the Pan-African Orogeny and may present a risk of current activities (Inger, Dione, Jarosewich-Holder, & Olivry, 2006), (El Abbass et al., 1993). Certain portions of the region are superimposed on an area of positive gravity anomalies (Svensen et al., 2003). The parallelism between the gravity and structural directions is particularly marked between Tombouctou, Faguibine and the Nara Trench (Chudeau, 2018; Zwarts, Van Beukering, Kone, & Wymenga, 2005). In April 2001, has been receiving reports about increased thermal activity in Tombouctou area (Svensen et al., 2003). Hot fumaroles and magmatic rocks whose the upper part of a hydrothermal system has been observed and considered as volcanic activity (Villatte, 1973). This study focuses specifically on multitemporal analysis of LST by a set of processes while also

emphasizing the variation of LST values under the influence of different sectors.

The authors appreciate the reviewer comments. As suggested by the reviewer, the data used are MODIS (MOD11A1) and have been well described in the revised manuscript as follows:

Generally, the MODIS instrument is operating on both the Terra and Aqua spacecraft. it has a viewing swath width of 2 330km and views the entire surface of the earth every one to two days Its detectors measure 36 spectral bands and it acquires data at three spatial resolution: 250m, 500m, and 1000m (LPDAAC (Land Processes Distributed Active Archive Center), 2018). Specifically, the MOD11A1_L3 version 6 product provides daily, per-pixel land surface temperature (LST) in a 1200 x 1200 kilometer grid in the Sinusoidal projection. The exact grid size at 1km spatial resolution is 0.928km by 0.928km. Using the generalized divided window algorithm under clear sky conditions defined in MOD35 (at a confidence> = 95% on land <= 2000 m or> = 66% on land> 2000 m, and with confidence> = 66% on the lakes), extracting the LST pixel values from each pellet are used to generate the daily product MOD11A1 LST C6. Through clear sky, this construction maps all valid LST values on the grids of the sinusoidal projection while using the average of the LST values of the superimposed pixels in each grid with superimposed areas by weight (Zhengming Wan, 2007, 2013). In clear skies, several observations are made to some pixels when the latitude is 30 degrees. Thus the generalized split-window algorithm retrieves the MOD11A1_L3 through the expression (Z. Wan & Dozier, 1996; Zhengming, 1999):

Equ1 Ts represents the LST T31 and T32 are band 31 and 32 of MODIS brightness temperature $\varepsilon 31$ and $\varepsilon 32$ are band 31 and 32 of MODIS surface emissivity that vary within a land cover type (crop lands may have different soils and crops in variable coverage); A MODIS pixel may cover several 1km grids with different land cover types (Z. Wan & Zhang, Y., Zhang, Q., & Li, 2004). C, A1, A2, A3, B1, B2 ,and B3 are the coefficients of regression and depend on viewing zenith angle (in range of 0-65°) The linear regression allows to establish the tables representing the multidimensional
look-up. The algorithm contains coefficients that also depend on the surface temperature ranges of air and water vapor of the MOD11A1_L3 column. These coefficients in addition to the zenith angle to improve the recovery accuracy of the LST are also incorporated (Z. Wan & Zhang, Y., Zhang, Q., & Li, 2004; Z Wan, Wang, & Li, 2004). Equ 2 LSTc Land Surface Temperature in Celsius degree DN Digital number represents the SDS (scientific data sets) data in uint16 which multiplied by the scale factor 0.02 gives a value in the range of 150-1310.7K (Zhengming Wan, 2013). The archived images have been selected from databases of the website "earth explorer" in the way that more than 95% of the study area is a good quality of visualization (Survey, 2005; USGS United States Geological Survey, 2016; Zhengming Wan, 2007). The raw data MOD11A1V6 with 0.928 kilometers of the spatial resolution has like unit the "uint16" which represents the multiplier of the scale factor (0.02). The value of this ratio entity ("DN x 0.02") is then expressed in Kelvin degree while situating it in the range of 150-1310.7K (Zhengming Wan, 2013). Characteristics of the effective calibration parameters of Scientific Data Sets (SDS) are expressed in the below Table A (Zhengming Wan, 2007). The used images MODIS MOD11A1V6 are selected over a period of eighteen years (2000 to 2017).

The authors appreciate the reviewer comments. -As suggested by the reviewer, the methodological process is clearly described in the manuscript as below. The used images MODIS MOD11A1V6 are selected over a period of eighteen years (2000 to 2017). The used method presents 3 separate parts whose preprocessing includes the data clipping with the study area outline followed by data screening into Kelvin degree, then a conversion in Celsius degree using the equation 2. The "equal Interval" method has allowed processing each image into 5 classes of temperature, due to the non-evolution of the LST distribution beyond 5 slices. In other words, the threshold of the variation of the classes on which is well detailed the distribution of the LST is limited to 5 classes (figure 5). As shown in Figure 6, the 3D modeling of the LST is strongly influenced by the geological occupancy. Thus, different transition sectors of the temperature slices were identified while determining the transfer rates. Based on
the decision tree technic (Bogaert, Ceulemans, & Salvador-Van Eysenrode, 2004), different sequence types were observed for each of the transitions in order to calculate the rate of evolution of the sequence types. - As suggested by the examiner. The "Characterization of the LST modeling in function of geology" section explains the influence of geology in the variation of LST through modeling. And have been well described in the revised manuscript as follows The characteristic of the LST shape is further builted as based on the continuous and smooth modelling. Figure 6 illustrates the 3D format of the Mean of 18 years. The morphology of the modelling is well described over the area. The diversified mean LTS values "from 19.21°C to 48.94°C" show from the geology and landscape configuration an appreciable demonstration. The morphological typical shapes such as concave, ridge, and flat part are illustrated in Figure 6 by taking the mean as an example. The three regions has revealed that the LST values of hottest areas are represented by the mountain ridges (the highest), while the lowest temperatures are illustrated by concave cavities, and followed by the mean temperatures which are illustrated by flat shapes (Dembélé & Ye, 2017; Mahe, Orange, Mariko, & Bricquet, 2011; Maiga. H, Marie. J, Morand. P, N'Djim. H, 2007). Accordingly the modelling, the first region performs as a concave shape as a water area surrounded by wet area included between "19.21°C to 31.10°C" in the recent alluvions (Dembélé & Ye, 2017). Also mainly as a semi-dry area, the second region demonstrates as a flat part between "31.10°C to 43.00°C" into the arrangement of precambrian and continental terminal formations at southwest and at the East of recent alluvions. The third region reveals a ridge morphology as it is a mixed Dunes, sandy, clays, and laterite at the northwest of the study area include between "43.00°C to 48.94°C". Therefore, the diversification of the geology also influences the values of the LST as the modeling shows.

The authors appreciate the reviewer comments. Accordingly as suggested, the color bars as well the "high" and "low" of is marked in the figure 3 and the figure 6.

The authors appreciate the reviewer comments. Accordingly as suggested the reviewer, the relevance of work has been better discussed in the conclusions section as follow.

One of the studies on the environment or climate is the variation of the LST. The state of the environment having undergone various changes of temperature at its surface level provides different LST values. These LST values generated from MODIS data require enhanced analysis to predict the progress of global warming on different sectors over time. Based on MOD11A1 data, LST values are examined over an 18-year period (2000 to 2017). The spread of the LST dynamics highlights different temperature slices whose threshold did not exceed 5 slices. That is to say beyond 5 slices, no longer evolves the distribution of LST in the study area. This process of processing made it possible to limit certain errors related to the multitemporal analysis of the images by the lightening in slices of temperatures. Over an eighteen-year interval, the strong stagnation of certain temperature slices shows a slight increase compared to other slices. The dynamics in each of the years are not able to dominate global interannual dynamics. Technically, the core of this study is to analyze the frequency of variation of the LST as well as the factors influencing the LTS values. After image processing, a variation of the LST was found as much as at the level of the temperature slices, as well as at the level of the different years (interannual). The LST has an increasing value from the southern part to the northern part. The variation of the LST during the 18 years shows an annual increase of the temperature of 0.24 ° C. Throughout the means of the temperature slices, the smallest spaces are occupied by the maximum and the minimum of the temperature slices "43.00 ° C to 48.94 ° C and 19.21 ° C to 25.15 ° C" with 2.02% and 4.02 % respective, while the largest area representing 64.59% is occupied by the mean of the slices "37.05 ° C to 43.00 ° C." And the LST values are influenced by some of the hottest surfaces, among which are the faguibine system. in addition, at the environmental scale the surfaces of the first two maximum slices "37.05 ° C to 43.00 ° C and 43.00 ° C to 48.94 ° C" indicate respective regressions of -0.64% and - 1.24% per annum and directed towards the North, while the average area of the slice "31.10 ° C at 37.05 ° C" undergoes a strong progression of 1.74% per year and

the surfaces two minimum slices "19, 21 ° C at 25.15 ° C and 25.15 ° C at 31.10 ° C " have slight increase rates of 0.13% and 0.20% respectively. In addition, the study area is instituted of a geological set that plays an important role in LST modeling. In fact, different forms have characterized the LST modeling for the existing geology. They are identified as follows: the concave shape, the ridge and the flat shape. The concave shape represents the lowest temperature slice "19.21 ° C to 31.10 ° C" observed on the watercourse (Niger river) and its surroundings. The flat shape represents the mean slice "31.10 ° C to 43.00 ° C" in the arrangement of the Precambrian and continental terminal formations scattered on the southwest and southeast coasts of the study area. And finally the maximum temperature range "43.00 ° C to 48.94 ° C" is represented by the ridge shape and is located in the mixture of dunes, sands, clays and laterite. The modeling reports an overview of the LST variation characterized by geology. In other words, geology has a part to affect the LST values. The variation of the LST observed on all the images corresponds to the rate of temperatures transited from one year to the next. Globally, the period from 2000-2001 to 2016-2017 was characterized by three main types of temperature dynamics: a stability (50.60% of the space), a dynamics of narrowing of low temperature (46.37% of the space), and an opening of High temperatures (2.63%) expressing in varying degrees. Many of the means slices (third and fourth slices) of the different years have their major portion remaining intact in subsequent years. Between the maximum slices, the lowest transition rates are observed, followed by the minimum slices (first and second slices). In the set, the quantization of the interannual transition of temperature slices shows a dominance of space by means temperature slices with a small variation of the LST. And the maximum temperature slices showing the low transition rates define a large interannual increase of the LST. The transited temperature ranges occupied the surface in a particular way. This way of occupation by the slices is determined by the spatial transformation processes (STP) in order to write the evolution status of the LST. Based on the technic of the decision tree, a sequence of two major changes (creation and attrition) have invaded all the different interannual temperature ranges. Of those, the maximum interannual temperature

ranges have suffered more decrease in the number of spots than creation, while on the four remaining temperature bands, have been observed the maximum of the creation of new spots. Hence in the set, 53.33% of spots creation against with 46.66% of spot attrition constitute the STP. Thus the disappearance of the spots (attrition) on the means of the maximum temperature slices from 7.78% to 12.22% causing a decrease of its surface up to 4.02%.

The authors appreciate the reviewer comments. As suggested by the reviewer, the set of the references of the manuscript has been revised and correctly cited in the text (, lines 29-30).

[Figure]

$$T_S = C + \left(A_1 + A_2 \frac{1-\varepsilon}{\varepsilon} + A_3 \frac{\Delta\varepsilon}{\varepsilon^2}\right)\frac{T_{31}+T_{32}}{2} + \left(B_1 + B_2 \frac{1-\varepsilon}{\varepsilon} + B_3 \frac{\Delta\varepsilon}{\varepsilon^2}\right)\frac{T_{31}-T_{32}}{2} \qquad 1$$

$$\varepsilon = 0.5\,(\varepsilon_{31} + \varepsilon_{32}) \ \text{ and } \ \Delta\varepsilon = (\varepsilon_{31} - \varepsilon_{32})$$

$$LST_c = (DN * 0.02) - 273.15 \qquad 2$$

**Fig. 1.**

Table A. SDSs in the MOD11A1 product

| SDS Name | Long Name | Number Type | Unit | Valid Slice | Fill Value | Scale factor | Add offset |
|---|---|---|---|---|---|---|---|
| LST_Day_1km | Daily daytime 1km grid Land-surface Temperature | Unit16 | Kelvin | 07500-65535 | 0 | 0.02 | 0 |
| QC-Day | Quality control for daytime LST and Emissivity | Unit18 | none | 0-255 | 0 | NA | NA |
| Day_view_time | (local solar) time of daytime land-surface temperature observation | Unit18 | hrs | 0-240 | 0 | 0.1 | 0 |
| Day_view_angle | View Zenith angle of daytime Land-surface Temperature | Unit18 | degree | 0-130 | 255 | 1.0 | -65.0 |
| LST_Night_1km | Daily nighttime 1km grid Land-surface temperature | Unit16 | Kelvin | 7500-65563 | 0 | 0.02 | 0.0 |
| QC_Night | Quality control for nighttime LST and emissivity | Unit8 | none | 0-255 | 0 | NA | NA |
| Night_view_time | (local solar) time of nighttime land-surface temperature observation | Unit8 | hrs | 0-240 | 0 | 0.1 | 0 |
| Night_view_angle | View Zenith angle of nighttime Land-surface Temperature | Unit8 | degree | 0-130 | 255 | 1.0 | -65.0 |
| Emis_31 | Band 31 emissivity | Unit8 | none | 1-255 | 0 | 0.002 | 0.49 |
| Emiss_32 | Band 32 emissivity | Unit8 | none | 1-255 | 0 | 0.002 | 0 |
| Clays_day_cov | Day clear-sky coverage | Unit16 | none | 0-65535 | 0 | 0.0005 | 0 |
| Clay_night_cov | Night clear-sky coverage | Unit16 | none | 0-65535 | 0 | 0.0005 | 0 |

**Fig. 2.**

[Figure]

Figure 3. Land Surface Temperature (LST) change from 2000 to 2017

**Fig. 3.**

**Figure 6.** Modelling of the mean LST

**Fig. 4.**

---

## Author Comment (AC2) · 14 Dec 2018

1) The authors thank the reviewer for thoroughly reviewing our manuscript, providing valuable suggestions to improve the manuscript. It convenient that this study focuses on the analysis of the temporal variation of LST values in order to estimate the spatiotemporal-effects on environment during 18 years. Accordingly, a common response has been prepared for this part concerning the objective and other data sources such as litholgy, hydrological, and Rainfall.

The MODIS image is a good indicator of Land surface temperature (LST) on the interface analysis in order to characterize areas. The latest decades has registered the

attraction of much attention through the LST at large-scale (Roy et al., 2014; Vlassova, Pérez-Cabello, Mimbrero, Llovería, & García-Martín, 2014; Wang, Liang, & Meyers, 2008; Zorer et al., 2013) and the advancement of earth and environmental sciences in the desertification monitoring so as in the monitoring of LST evolution (Hillger & Clark, 2002a, 2002b). One of the major driving forces causing many extremes of soil anomalies whose in recent years that many regions have undergone is the terrestrial global warming (C et al., 2013; Committee, 2015; Tol, 2009). Based on meteorological stations data, many studies have quantified the global temperature of the earth (Coumou, Robinson, & Rahmstorf, 2012; Frangou, Ladle, Malhado, & Whittaker, 2010; Rahmstorf & Coumou, 2011). Then several phenomena and climate extreme have been recorded during 2011–2015 and the probability increased by a factor of ten or more, in the case of some extreme high temperatures (World Meteorological Organization, 2016a)(World Meteorological Organization, 2016b). Only temperature anomalies have concerned most analysis(Report, 2018; Weber et al., 2014). The Inner Delta of Niger (IDN), in central of Mali, through desert regions represents one of the hot areas of the African plate (Friese, 2010; Mao, K et al., 2017). Inner Delta of Niger, is part of a large geological structure (Continental Terminal, Precambrian, Quaternary) that have played several roles since the Pan-African Orogeny and may present a risk of current activities (Inger, Dione, Jarosewich-Holder, & Olivry, 2006), (El Abbass et al., 1993). Certain portions of the region are superimposed on an area of positive gravity anomalies (Svensen et al., 2003). The parallelism between the gravity and structural directions is particularly marked between Tombouctou, Faguibine and the Nara Trench (Chudeau, 2018; Leo Zwarts, Van Beukering, Kone, & Wymenga, 2005). In April 2001, has been receiving reports about increased thermal activity in Tombouctou area (Svensen et al., 2003). Hot fumaroles and magmatic rocks whose the upper part of a hydrothermal system has been observed and considered as volcanic activity (Villatte, 1973). This study focuses specifically on multitemporal analysis of LST by a set of processes while also emphasizing the variation of LST values under the influence of different sectors as well its effects on environmental lithology, hydrological, Rainfall and others.

The climate variability and the hydrological regime It was calculated before 1923, the maximum flood levels from the relationship between the average monthly flow measured at Koulikoro in September. Based on daily measurements of the water level as shown the Figure 7, the Mopti area receives the peak of the flood in 1929, then Akka and Dire around 1957. From there, a gradual decrease in the water level until 1984 (year catastrophic dry) is observed, which reduced the flooded area to 7800 km2 against 22.000 km2 during the years that preceded. Between 1994 and 2012, the maximum floods that never reached the previous peaks, vary very randomly with slight regressions of the peak levels (Sangaré, S., Mahé, G., Paturel, J.-E., et BANWURA, Y, 2002; L. Zwarts & Hoekema, 2013). The dates of the maximum flood have returned to near the 70s at the beginning of the drought, but remain earlier than during the wetter years before 1970 (at least 1.5 months) (Zaré, 2015). The cumulative volume at the exit (outlet) of the Inner Delta in Diré region during the period from June 1, 2015 to April 30, 2016 is 27.79 109 m3. This volume is higher than that of 2014/2015 (25.74 109 m3) and lowest than that of 2013/2014 (32.87 109 m3) and that of statistical mean from 1924 to 2012 (30.38 109 m3) (Niger Basin Authority, 2016). The effect of terrestrial warming represents one of the potential factors of the decrease of the water level in the different sectors, it is also observed on the images and statistics of the LST through the progressive variation of its values.

LST and the geology The characteristic of the LST shape is further builted as based on the continuous and smooth modelling. Figure 6 illustrates the 3D format of the Mean of 18 years. The morphology of the modelling is well described over the geology (lithology) area (Dembélé & Ye, 2017; Mahe, Orange, Mariko, & Bricquet, 2011; Maiga. H, Marie. J, Morand. P, N'Djim. H, 2007). The diversified mean LTS values "from 19.21°C to 48.94°C" show from the geology and landscape configuration an appreciable demonstration. The morphological typical shapes such as concave, ridge, and flat part are illustrated in Figure 6 by taking the mean as an example. The three regions has revealed that the LST values of hottest areas are represented by the mountain ridges (the highest), while the lowest temperatures are illustrated by concave cavities, and followed by the mean temperatures which are illustrated by flat shapes. Accordingly the modelling, the first region performs as a concave shape as a water area surrounded by wet area included between "19.21°C to 31.10°C" in the recent alluviums (Figure 7) Also a semi-dry area, the second region demonstrates as a flat part between "31.10°C to 43.00°C" at southwest into the arrangement of Ancient alluvium and continental terminal formations, at South-east into the Precambrian and at a small part of north over the dunes and sandy area. The third region reveals a ridge morphology in a mixed Dunes, sandy, clays, and laterite at the northwest of the study area include between "43.00°C to 48.94°C". Therefore, the diversification of the geology also influences the values of the LST as the modeling.

LST, rainfall and geology This continuity phenomenon of terrestrial warming tends to increase the water resources needs by affecting of the different systems of aquifer existing in the study area. From the rainfall point of view, precipitation decreases have been observed since the end of the 1950s which has been exceptionally high, and then the last three decades of the period 1961 to 2004 (Ministere De L'equipement Et DesTransports-MALI, 2007; Zwart, 2010). The isohyets of the 1981-2010 normal (Figure 7b) are characterized in the Saharan and Sahelo-Saharan zone of Mali by a slight rise towards the North-East compared to the normal 1961 - 1990 (Figure 7a) (Gicresait, 2017). In the set, the influence of LST values caused by warming contributed to the variability of precipitation. Consisting of three major aquifer systems, the major part of the study area is occupied by the generalized aquifers consisted of the continental terminal/quaternary or Intercalary stratigraphy. They are characterized by the permeability of intergranular types and by a continuous aquifer. Consisting of the Precambrian tabular/folded stratigraphy representing the second big area, cracked aquifers have a low permeability. They are endowed with water reserves from other environment through deep cracks, so these aquifers receive more drillings than any others aquifer system. The superficial aquifers being recent formations is consisted only of the quaternary stratigraphy, then play an important infiltration role because of its non-consolidation. All of these aquifer systems today continue to receive more drilling

to compensate for the need for water resources due to the decrease or tarrissemnet of water sources (Lake Korientzé, Lake Débo, Lake Takara, Niangay Lake, Do Lake, Garou Lake, Haribongo Lake, Kabara Lake, Tanda Lake, Gouber Lake, Fati Lake, Horo Lake, Lake Tele, Kamango Lake, and Figuibine Lake) by the effect of warming up.

2) The authors thank the reviewer for thoroughly reviewing our manuscript, providing valuable suggestions to improve the manuscript. Therefore a language review has been made to the entire manuscript after all corrections addressed in the following paragraphs including the add of new source of data for disasters assessments. After an assessment of the - Trends and variations in minimum, maximum and mean temperatures observed, the impact on the hydrological regime by climate variability has been carried and evaluated. It was calculated before 1923, the maximum flood levels from the relationship between the average monthly flow measured at Koulikoro in September. Based on daily measurements of the water level as shown the Figure 7, the Mopti area receives the peak of the flood in 1929, then Akka and Dire around 1957. From there, a gradual decrease in the water level until 1984 (year catastrophic dry) is observed, which reduced the flooded area to 7800 km2 against 22.000 km2 during the years that preceded. Between 1994 and 2012, the maximum floods that never reached the previous peaks, vary very randomly with slight regressions of the peak levels (Sangaré, S., Mahé, G., Paturel, J.-E., et BANWURA, Y, 2002; L. Zwarts & Hoekema, 2013). The dates of the maximum flood have returned to near the 70s at the beginning of the drought, but remain earlier than during the wetter years before 1970 (at least 1.5 months) (Zaré, 2015). The cumulative volume at the exit (outlet) of the Inner Delta in Diré region during the period from June 1, 2015 to April 30, 2016 is 27.79 109 m3. This volume is higher than that of 2014/2015 (25.74 109 m3) and lowest than that of 2013/2014 (32.87 109 m3) and that of statistical mean from 1924 to 2012 (30.38 109 m3) (Niger Basin Authority, 2016). The effect of terrestrial warming represents one of the potential factors of the decrease of the water level in the different sectors, it is also observed on the images and statistics of the LST through the progressive variation of its values.

LST and the geology The characteristic of the LST shape is further builted as based on the continuous and smooth modelling. Figure 6 illustrates the 3D format of the Mean of 18 years. The morphology of the modelling is well described over the geology (lithology) area (Dembélé & Ye, 2017; Mahe, Orange, Mariko, & Bricquet, 2011; Maiga. H, Marie. J, Morand. P, N'Djim. H, 2007). The diversified mean LTS values "from 19.21°C to 48.94°C" show from the geology and landscape configuration an appreciable demonstration. The morphological typical shapes such as concave, ridge, and flat part are illustrated in Figure 6 by taking the mean as an example. The three regions has revealed that the LST values of hottest areas are represented by the mountain ridges (the highest), while the lowest temperatures are illustrated by concave cavities, and followed by the mean temperatures which are illustrated by flat shapes. Accordingly the modelling, the first region performs as a concave shape as a water area surrounded by wet area included between "19.21°C to 31.10°C" in the recent alluviums (Figure 7) Also a semi-dry area, the second region demonstrates as a flat part between "31.10°C to 43.00°C" at southwest into the arrangement of Ancient alluvium and continental terminal formations, at South-east into the Precambrian and at a small part of north over the dunes and sandy area. The third region reveals a ridge morphology in a mixed Dunes, sandy, clays, and laterite at the northwest of the study area include between "43.00°C to 48.94°C". Therefore, the diversification of the geology also influences the values of the LST as the modeling.

LST, rainfall and geology This continuity phenomenon of terrestrial warming tends to increase the water resources needs by affecting of the different systems of aquifer existing in the study area. From the rainfall point of view, precipitation decreases have been observed since the end of the 1950s which has been exceptionally high, and then the last three decades of the period 1961 to 2004 (Ministere De L'equipement Et DesTransports-MALI, 2007; Zwart, 2010). The isohyets of the 1981-2010 normal (Figure 7b) are characterized in the Saharan and Sahelo-Saharan zone of Mali by a slight rise towards the North-East compared to the normal 1961 - 1990 (Figure 7a) (Gicresait, 2017). In the set, the influence of LST values caused by warming contributed to the variability of precipitation. Consisting of three major aquifer systems, the major part of the study area is occupied by the generalized aquifers consisted of the continental terminal/quaternary or Intercalary stratigraphy. They are characterized by the permeability of intergranular types and by a continuous aquifer. Consisting of the Precambrian tabular/folded stratigraphy representing the second big area, cracked aquifers have a low permeability. They are endowed with water reserves from other environment through deep cracks, so these aquifers receive more drillings than any others aquifer system. The superficial aquifers being recent formations is consisted only of the quaternary stratigraphy, then play an important infiltration role because of its non-consolidation. All of these aquifer systems today continue to receive more drilling to compensate for the need for water resources due to the decrease or tarrissemnet of water sources (Lake Korientzé, Lake Débo, Lake Takara, Niangay Lake, Do Lake, Garou Lake, Haribongo Lake, Kabara Lake, Tanda Lake, Gouber Lake, Fati Lake, Horo Lake, Lake Tele, Kamango Lake, and Figuibine Lake) by the effect of warming up.

[Figure]

**Figure 7** The annual variation of the maximum water level in the Inner Niger Delta (L. Zwarts & Hoekema, 2013-Mali DNH data)

**Fig. 1.**

[Figure]

**Figure 8.** Modelling of the mean LST

**Fig. 2.**

[Figure]

Figure 9 The lithology

Fig. 3.

[Figure]

**Figure 10** Representation of the isohyets; (a) Normal 1961_1990; (b) Normal 1981_2010 (Gicresait Projet; Sahara and Sahel Observatory; 2017)

**Fig. 4.**

---

## Author Comment (AC3) · 21 Dec 2018

The authors thank the reviewer for thoroughly reviewing our manuscript, providing valuable suggestions to improve the manuscript. Therefore a language review has been made to the entire manuscript after all corrections addressed in the following paragraphs including the addition of the rest of the new data source for disaster assessment.

- Variation of groundwater levels in the face of climate change One of the priority axes in an analysis is to evaluate the climatic stress resulting in the lowering of the level of the aquifers and the increasing exploitation of the resource taking into account the external

climatic conditions. The lowering of the tablecloth level threatened by a significant acceleration in the views of global warming shows different vulnerable zones including those in danger. The attention to take care of these areas needs more oversight as they define priority areas for water management. Overall, the Ilullemeden-Taoudéni / Tanezrouft (ITAS) aquifer system remains very little to little vulnerable to more than 80% at lowering of piezometric levels (Figure 11) (Gicresait, 2017). Considering the statistics generated in the study area as shows the Figure 12, 33.82% of the area is considered not very vulnerable and are mostly located in the northwestern part of the study area. The vulnerable zones detected in the center in the south-north direction occupy 26.34%, then 24.94% of the surface is occupied by the highly vulnerable zones, most of which are in the east, and the moderately vulnerable zones occupy 5.44% of the surface. Finally, the areas highly vulnerable (0.62%) and areas with very low vulnerability (2.94%) are located respectively in the southwestern part and the eastern part of the study area. The area extending from the area Vulnerable to the highly vulnerable zone appear excessively exploited by the population as a result of global warming, whereas the less vulnerable areas and the areas of low vulnerability with less populated occupation appear to be sufficiently preserved from this vulnerability.
* * *
[Figure]

**Figure 11** Vulnerability of groundwater related to climate change in the Ilullemeden-Taoudéni / Tanezrouft aquifer system (ITAS) (Sahara and Sahel Observatory 2013)

**Fig. 1.**

[Figure]

**Figure 12** Statistics concerned by the study area

**Fig. 2.**